# Micelle-Containing Hydrogels and Their Applications in Biomedical Research

**DOI:** 10.3390/gels10070471

**Published:** 2024-07-17

**Authors:** Jinghua Wu, Huapeng Li, Nan Zhang, Qingfei Zheng

**Affiliations:** 1Department of Radiation Oncology, College of Medicine, The Ohio State University, Columbus, OH 43210, USA; 2Center for Cancer Metabolism, James Comprehensive Cancer Center, The Ohio State University, Columbus, OH 43210, USA; 3Molecular, Cellular, and Developmental Biology Graduate Program, The Ohio State University, Columbus, OH 43210, USA; 4Department of Biological Chemistry and Pharmacology, College of Medicine, The Ohio State University, Columbus, OH 43210, USA

**Keywords:** hydrogel, micelle, molecular network, stimuli-responsive material, drug delivery, biocompatibility

## Abstract

Hydrogels are one of the most commonly used materials in our daily lives, which possess crosslinked three-dimensional network structures and are capable of absorbing large amounts of fluid. Due to their outstanding properties, such as flexibility, tunability, and biocompatibility, hydrogels have been widely employed in biomedical research and clinics, especially in on-demand drug release. However, traditional hydrogels face various limitations, e.g., the delivery of hydrophobic drugs due to their highly hydrophilic interior environment. Therefore, micelle-containing hydrogels have been designed and developed, which possess both hydrophilic and hydrophobic microenvironments and enable the storage of diverse cargos. Based on the functionalities of micelles, these hydrogels can be classified into micelle-doped and chemically/physically crosslinked types, which were reported to be responsive to varied stimuli, including temperature, pH, irradiation, electrical signal, magnetic field, etc. Here, we summarize the research advances of micelle-containing hydrogels and provide perspectives on their applications in the biomedical field based on the recent studies from our own lab and others.

## 1. Introduction

Hydrogels are a widely used type of soft material constructed with crosslinked three-dimensional (3D) molecular networks. Due to their special structures, hydrogels are able to absorb a large amount of fluid and possess flexibility, softness, tunability, and biocompatibility [1]. Therefore, hydrogels are commonly employed in the biomedical field (Figure 1), such as drug delivery [2], biosensors [3], wound dressings [4], bioelectronic devices [5], contact lenses [6], etc. To construct the molecular networks within hydrogels, two major crosslinking approaches have been applied [7], i.e., chemical and physical crosslinking-based syntheses. Specifically, the chemical crosslinking strategy allows for the formation of covalent bonds between hydrophilic polymers within the molecular networks during gelation, which makes the resulting hydrogels more difficult to be biodegradable, depending on the types of formed covalent bonds [8]. Therefore, chemically crosslinked hydrogels have been widely applied as artificial implants [9]. Conversely, the physically crosslinked hydrogels (also referred to as supramolecular hydrogels) are established via non-covalent cohesion forces, such as electrostatic interactions, ligand–metal coordination, π–π stacking, hydrogen bonding, etc [10]. Thus, the physically crosslinked hydrogels possess highly dynamic features, such as self-healing properties and injectability [11]. Overall, due to their unique structures and outstanding biocompatibility, both covalently crosslinked and supramolecular hydrogels have become the most commonly employed materials in biomedical research and clinic [12], especially for controlled drug release [13].

However, traditional hydrogels face obvious limitations in drug delivery, particularly for drugs with low water solubility [14]. It has been well established that canonical hydrogel matrices are proficient at immobilizing water-soluble drugs [15], while they are not ideal for the storage of hydrophobic drugs due to their highly hydrophilic internal environment [16]. To overcome this drawback, a variety of modifications to hydrogels have been developed and applied. One key method is to introduce micelles into the hydrogel systems (Figure 2), which can make them possess a dual microenvironment of both hydrophilicity and hydrophobicity [17,18]. A micelle is a group of surfactant molecules aggregated in an aqueous solution, where the water-attracting heads face outward, and the water-repelling tails are hidden inside, forming a colloidal suspension [19]. Due to the existence of hydrophobic cavities within micelles, micelle-containing hydrogels are able to store cargos with poor water solubility for the subsequent controlled release [20]. In these hydrogel systems, micelles can serve as “containers” for various hydrophobic substances, thereby enhancing the capability of corresponding hydrogels to encapsulate and deliver diverse contents [21].

In addition to providing hydrophobic microenvironments, the micelles have also been utilized for the construction of 3D molecular networks of hydrogels. This type of hydrogel, of which micelles are employed as crosslinking agents, is known as a micelle-crosslinked hydrogel [22]. The micelles can be integrated into hydrogels together with hydrophilic polymers through two major crosslinking approaches, i.e., chemical and physical crosslinking (Figure 2), resulting in functional materials with distinct properties [23]. Specifically, for chemically crosslinked micelle-containing hydrogels, the micelle surfaces typically feature reactive moieties, such as C-C double bonds, that can participate in free-radical polymerization. Under specific conditions, these reactive moieties within micelles react with the polymers, resulting in the formation of covalently crosslinked rigid molecular networks [24,25]. In contrast, the physically crosslinked micelle-containing hydrogels are formed via supramolecular forces, which feature transient crosslinking formed by the associations between polymer chains and micelles through non-covalent bonds or dynamic reversible covalent bonds [26]. The reversible nature of these physically crosslinked networks imparts unique material properties, such as outstanding stretchability, toughness, adhesion, injectability, and self-healing [27,28]. Collectively, micelle-crosslinked hydrogels possess outstanding qualities and good performance, which are attributed to their unique molecular network structures and thus have been widely applied in biomedical research.

In this review, we will summarize the design and preparation of micelle-containing hydrogels. Representative examples will be discussed (Table 1), as well as the responses of these hydrogels to diverse stimuli. Finally, we will highlight the applications of micelle-containing hydrogels in the biomedical field and provide future perspectives, with a particular focus on their advantages in drug delivery for disease therapeutics.

## 2. Micelle-Containing Hydrogels

Due to their instability, low local concentrations, and other issues of micelles in complex biological environments, directly using micelles as carriers for the sustained delivery of therapeutic agents remains challenging [43]. Furthermore, loading micelles into hydrogels can effectively address these issues for a broader application range in biomedical research and clinics. Based on the functionality of micelles within hydrogels, which either contribute to the construction of molecular networks or not, micelle-containing hydrogels can be briefly classified into micelle-doped and micelle-crosslinked ones (Figure 2). The molecular networks of the former type are constructed regardless of the micelles, while the gel formation of the latter type heavily relies on the existence of micelles that maintain the crosslinking forces.

### 2.1. Micelle-Doped Hydrogels

Micelles are constructed by amphipathic molecules [44] and easily incorporated into hydrophilic matrices to form micelle-doped hydrogels [45]. For instance, the Madhavamenon group reported a type of hybrid hydrogel, which was formed by the incorporation of resveratrol micelles into a fenugreek galactomannan hydrogel [29]. In this hybrid system, fenugreek galactomannan was used to construct the hydrogel matrix, and the resveratrol micelles were uniformly dispersed into the molecular network through homogenization. The resulting mixture was then dehydrated under vacuum to obtain a micelle–hydrogel composite in a powder form (RF-20). Under gastrointestinal conditions, RF-20 could absorb water and swell to reform a soft hydrogel, and meanwhile, resveratrol micelles were released in a sustained manner for better absorption. Notably, the experimental results showed that RF-20 could significantly increase the absorption of free resveratrol in the human body, with enhanced pharmacokinetic properties, e.g., a much higher maximum plasma concentration and a longer half-life, compared to unformulated resveratrol. Moreover, RF-20 significantly improved the total bioavailability of resveratrol, which was over ten times better than that of unformulated resveratrol. In general, this micelle-doped hydrogel system demonstrates significant potential in enhancing the bioavailability of resveratrol, suggesting a promising methodology to improve the therapeutic efficacy of lipophilic bioactive compounds.

In some micelle-doped hydrogels, the micelles are not pre-prepared but formed during the gelation of materials [46]. For example, back in 1995, the Okano group designed and developed a thermosensitive hydrogel with a comb structure, where poly(*N*-isopropylacrylamide) (PIPAAm) chains were grafted onto the crosslinked molecular networks [30]. Unlike conventional PIPAAm-crosslinked hydrogels, the grafted PIPAAm chains in this system possessed mobile ends, allowing the materials to transform from an expanded (hydrated) form to a compact (dehydrated) form more readily when the environmental temperature increased. This collapse occurred before the PIPAAm network began to shrink due to the mobility of grafted chains. The dehydration of these grafted chains created hydrophobic clusters (i.e., self-formed micelles), which enhanced the aggregation of crosslinked chains and facilitated a rapid de-swelling and shrinking of the hydrogel. Notably, the comb-type grafted PIPAAm hydrogel could reach equilibrium in 20 min when the temperature was increased from 10 °C to 40 °C, whereas the same process usually took more than a month for conventional gels. For hydrogel-based smart materials and actuators, the rapid de-swelling and shrinking responses can be utilized to develop fast-acting sensors and control devices [47]. Overall, the amphipathic nature and hydrophobic cavity-containing structures of micelles enable micelle-doped hydrogels to possess complementary advantages in comparison with canonical hydrogels, which can be widely applied in biomedical research, including sustained drug release, tissue engineering, etc.

### 2.2. Micelle-Crosslinked Hydrogels

In another type of micelle-containing hydrogel, micelles participate in the crosslinking and construction of molecular networks, which have direct impacts on the structures and properties of the final materials. It has been shown that micelle-crosslinked hydrogels can offer significant improvements over traditional hydrogels with their enhanced mechanical and self-healing properties [48,49]. These advancements broaden the application scope of hydrogels [50], addressing the many obvious limitations associated with conventional hydrogels. In fact, compared with biological tissues, such as skin, muscle, and blood vessels, traditional hydrogels tend to be brittle and display inadequate mechanical characteristics [51]. These shortcomings have largely limited the applications of hydrogels in biomedical research. Therefore, developing novel micelle-crosslinked hydrogels with superior mechanical properties, including ultra-high toughness, stretchability, and elasticity, is of paramount importance in biomedical engineering.

#### 2.2.1. Chemically Crosslinked Micelle Hydrogels

In polymer chemistry, micellar copolymerization is a simple and effective technique to embed hydrophobic domains within micelles [52], which can act as multifunctional crosslinkers to form molecular networks of hydrogels [53]. This technique has been frequently used to prepare tough hydrogels with unique mechanical properties [54].

For example, the Selb group reported a type of micellar copolymerization-based hydrogel, which was a bioerodible material constructed by photopolymerized polyethylene glycol (PEG) and α-hydroxy acids. [17]. In the molecular network of this hydrogel, the monomer consisted of a PEG central block and α-hydroxy acid oligomers as extending segments with acrylate groups at the ends. Due to the amphiphilic nature of PEG (hydrophilic) and α-hydroxy acids (hydrophobic), these structures spontaneously formed micelles in aqueous solutions. The subsequent photopolymerization led to crosslinking between the monomers, resulting in a 3D network structure within the hydrogel. Notably, the acrylate groups in micelles aggregated, making the polymerization reaction more efficient and thus forming a more uniform crosslinked network. Benefiting from the biocompatibility and degradability of the PEG component, along with the hydrolyzable nature of α-hydroxy acids, this chemically crosslinked micelle hydrogel could be successfully applied in drug delivery and tissue engineering.

In 2013, the Guo Lab developed a novel micellar crosslinking copolymerization method using amphiphilic polyurethane (PU) macromonomers that could self-assemble into micelles, thus eliminating the need for small molecular surfactants [31,32]. Specifically, polyurethane macromonomers (PUI and PUII) were synthesized using polyethylene glycol (PEG), isocyanate (IPDI), and acrylate (AOI) units. These amphiphilic macromonomers self-assemble into micelles in aqueous solutions, where the hydrophobic IPDI units form the core and the hydrophilic PEG chains construct the shell. Hydrogels were then formed by free-radical copolymerization of acrylamide (AAm) with micellar PUI or PUII macromonomers under ultraviolet (UV) light using α-ketoglutaric acid (α-KA) as the photoinitiator. In this hydrogel system, the micelles function as multifunctional crosslinkers, providing both physical and chemical crosslinking within the molecular network. This approach results in highly stretchable and resilient hydrogels with tunable mechanical properties and over 96% resilience at a 400% strain.

Another example is the Pluronic F127 diacrylate (F127DA)-based micelle hydrogels, which were synthesized by acrylating the end hydroxyl groups of Pluronic F127 using acryloyl chloride and triethylamine. F127DA is self-assembled in water to form micelles with vinyl groups on their surfaces. The Fu group and Yang group have utilized these F127DA-based micelles as crosslinking points by copolymerizing them with acrylamide (AAm) and N-acryloyl-6-aminocaproic acid (AACA) monomers to enhance the molecular network structures of the hydrogels and thus successfully improved their mechanical properties and toughness [55,56,57,58].

In 2019, the Yang group developed a micelle-crosslinked hydrogel by incorporating alkyl chain-protected ureido pyrimidinone (UPy) moieties into segmented copolymer backbones [33]. This study proposed a novel strategy based on strong and high-density micelle crosslinking for the preparation of non-swelling and tough supramolecular hydrogels. In this approach, micelle crosslinkers (NH_2_O-PEG-C8-UPy-C8-PEG-ONH_2_) were chemically crosslinked with aldehyde-terminated polyethylene glycol (OHC-PEG-CHO) at 37 °C through oxime bonds. This crosslinking reaction was highly rapid and efficient, forming a hydrogel network with a high crosslinking density. Future experimental results demonstrated that the C8-ONH_2_ micelle-crosslinked hydrogel exhibited excellent mechanical properties, with a compressive strength reaching 4 MPa and a maintained elasticity at high stretch ratios. Additionally, the C8-ONH_2_ micelle-crosslinked hydrogel showed almost no swelling after being immersed in phosphate-buffered saline (PBS) for one week, whereas the conventional hydrogels exhibited significant swelling and even complete degradation. The high-strength, non-swelling, and self-recoverable hydrogels prepared by using this strategy have huge potential for future biomedical applications.

In addition, a special micelle-crosslinked hydrogel with strain-stiffening properties was reported recently. The Huo group synthesized core-crosslinked micelles with adjustable intramicellar crosslinking densities and customizable chemical compositions using polymerization-induced self-assembly at high concentrations [59]. This work demonstrated that intramicellar crosslinking could effectively enhance the strain-stiffening properties of micelle-crosslinked hydrogel networks, allowing for the decoupled regulation of fracture stress and Young’s modulus, which created photoresponsive hydrogels with tunable mechanical properties and provided new insights into designing adaptive and tough hydrogels.

#### 2.2.2. Physically Crosslinked Micelle Hydrogels

Although chemically micelle-crosslinked hydrogels have been developed to address the mechanical limitations of canonical hydrogels through covalent crosslinking, this approach has resulted in relatively poor environmental sensitivity and biodegradability [60,61]. Additionally, during the crosslinking process, micelles undergo chemical reactions with polymer chains, which may have negative impacts on the encapsulated contents. Physically crosslinked hydrogels have better reversible properties due to the absence of covalent bonds, which makes up for the shortcomings of chemical crosslinking. In comparison, physically crosslinked micelle hydrogels, due to the lack of covalent bonds, exhibit superior reversibility without the limitations associated with chemical crosslinking [62].

For example, the Fu group used unmodified Pluronic 127 (F127) to form a type of non-swelling, super-tough, self-healing, and multi-responsive polymer hydrogel (PHFGs) utilizing micellar crosslinking with Pluronic F127 micelles [34]. In this hydrogel system, F127 micelles formed extensive hydrogen bonding and hydrophobic associations with poly(2-hydroxyethyl methacrylate) (pHEMA) chains, leading to a strong polymer network. The resulting hydrogels exhibited excellent mechanical properties, negligible swelling, self-healing ability, and responsiveness to diverse environmental stimuli (such as pH changes), making them suitable for smart switches and shape memory applications. Specifically, upon the addition of acid to this hydrogel, the hydrogen bonds were disrupted, thereby causing the material to swell and become transparent. As expected, when a base was added, the hydrogel reverted to its original state. Notably, the PHFGs demonstrated rapid self-healing under acidic conditions, attributed to the formation of new hydrogen bonds and the activation of polymer chain diffusion. Moreover, the PHFGs exhibited shape memory effects with changes in the water content, enabling them to be fixed into temporary shapes and recover to their original form upon water stimulation.

In 2023, the Yoo group developed a new type of hydrogel by incorporating L-glutamic acid (LGA) into hydrophobically crosslinked polyacrylamide (PAmm) chains, where the surfactant cetyldimethylethylammonium bromide (CDAB) was employed to introduce cationic micelles [35]. The introduction of LGA facilitated dynamic non-covalent crosslinking through electrostatic interactions and hydrogen bonding, acting as a connecting bridge between the micelle–micelle and micelle–polymer chains. This design significantly enhanced the mechanical properties of the material, including high extensibility (1650%), high toughness (740 kJ/m^3^), and a relatively high Young’s modulus (1.65 kPa) by increasing the crosslinking density through non-covalent interactions. The resulting hydrogels demonstrated rapid self-recovery and remarkable fatigue resistance, maintaining their mechanical properties after multiple deformations. In addition, the hydrogels in this research exhibited a high strain sensitivity, which was capable of accurately monitoring a wide range of human movements and subtle physiological signals. Collectively, this novel micelle–micelle crosslinking method offers new possibilities for manufacturing flexible hydrogel-based sensors with adjustable mechanical properties and reliable durability.

In 2023, the Zhang group reported a triple crosslinked micelle hydrogel, where the PEG-polylactic acid (PEG-PLA)-based micelles interacted with polymer chains and networks by acting as multifunctional crosslinkers, providing physical crosslinking points through hydrophobic interactions [36]. These supramolecular interactions enhanced the mechanical properties of the hydrogel, contributed to its self-healing capability, and enabled its application for sustained drug release. Therefore, the release duration of latanoprost and timolol was extended to 28 days, with the intraocular pressure-lowering effect in a rabbit model demonstrating an efficacy over five times higher than that of regular eye drops. These micelle hydrogel-based lacrimal sac implants could effectively reduce intraocular pressure and exhibited good safety and tissue compatibility, which have great potential for long-term and non-invasive treatment of glaucoma.

In general, physically crosslinked micelle hydrogels exhibit rapid association and dissociation rates due to their non-covalent interactions, such as hydrogen bonding, electrostatic interactions, hydrophobic interactions, and van der Waals forces, which contribute to the construction of molecular networks within hydrogels [63]. These dynamic supramolecular interactions enable the resulting hydrogels to serve as smart carriers for delivering versatile medicinal agents or as matrices for the repair and regeneration of organs and tissues in the human body [64]. Such dynamic crosslinking systems often result in hydrogel materials with high stretchability and other excellent mechanical properties, as well as adhesive properties [65]. These micelle hydrogels can actively rearrange themselves by remodeling, reshaping, and adapting to environmental stimuli, showcasing their diverse stimuli-responsiveness [66]. Overall, micelle-crosslinked hydrogels hold great promise for a wide range of biomedical applications, such as serving as drug delivery systems, tissue engineering scaffolds, wound dressings, and smart biomaterials that can adapt to complex environmental stimuli [67].

## 3. Stimuli-Responsive Micelle Hydrogels and Applications

In addition to the stimuli-responsive properties of polymer chains within their molecular networks, micelle-containing hydrogels possess more diverse stimuli-responsiveness, attributed to the existence of micelles that can also respond to environmental changes (Figure 3). For instance, the Duvall group developed a reactive oxygen species (ROS)-degradable and thermoresponsive micelle hydrogel, which possessed tunable mechanical properties and effective drug delivery capabilities (Figure 3A) [68]. Specifically, poly(*N*-isopropylacrylamide) (PNIPAAM) exhibits a lower critical solution temperature (LCST) of around 32 °C. Below the LCST, the PNIPAAM block is hydrophilic, thereby stabilizing the micelles in aqueous solution. When the temperature rises above the LCST (e.g., to body temperature at 37 °C), the PNIPAAM blocks hydrophilic to hydrophobic transitions. Additionally, this micelle hydrogel exhibited good compatibility with cells in vitro and demonstrated protective effects against ROS-mediated cell death. In a mouse model, the subcutaneous injection of PPS-b-PDMA-b-PNIPAAM polymer solution formed a stable hydrogel that could locally release the model drug, Nile Red, in a sustained fashion for up to two weeks. Collectively, this novel thermoresponsive micelle-containing hydrogel showcased excellent mechanical properties, degradation characteristics, and drug-release capabilities.

Recently, our lab developed a novel pH-responsive and recyclable micelle hydrogel [37]. This hydrogel system could sense environmental pH changes to control the reversible disassembly and reassembly of their molecular networks, which were constructed via one-pot radical polymerization, featuring dynamic hydrophobic interactions. Specifically, at a low pH, sodium dodecyl sulfate (SDS) micelles disassemble due to protonation and changes in the critical micelle concentration (CMC), thus breaking the hydrogel network. When the pH returns to neutral, adding fresh SDS and mineral salts allows the micelles to reassemble, thereby reforming the hydrogel. This pH-responsive behavior enables dynamic micelle–polymer interactions, where different mineral salts (e.g., NaHS, NaN_3_, and NaNO_2_) could play crucial roles in stabilizing micelles and reconstructing hydrophobic interactions.

Our lab also designed and developed an electro-responsive and dynamic micelle hydrogel with robust mechanical properties and precise spatiotemporal resolution (Figure 3C) [38]. This hydrogel system was designed and successfully developed based on the direct current voltage (DCV)-induced rearrangement of SDS micelles. Upon applying a direct current voltage, the SDS micelles within the hydrogel matrix rearranged, thus altering the mesh size of the hydrogel network. Notably, these hydrogel materials possessed high extensibility (>6000%) and high toughness (507 J/m^2^), exhibiting strong mechanical properties. Additionally, they featured self-healing and processability, which were reusable under dynamic conditions. The experimental results also indicated that after DCV treatment, a highly hydrophobic antibiotic, thiostrepton (TSR), which was encapsulated within the hydrophobic cavities of SDS micelles, could be released from the hydrogel, demonstrating its capability for on-demand drug delivery. The follow-up antibacterial experiments showed that TSR-loaded micelle hydrogels significantly inhibited the growth of methicillin-resistant *Staphylococcus aureus* (MRSA) after DCV loading. The model experiments conducted on porcine skin further demonstrated the potential of these micelle-containing hydrogels as wearable devices and bioelectronics in practical applications.

In 2021, the Sun group developed a type of micelle-containing hydrogel that combined excellent toughness, self-healing ability, and photoelectronic responsiveness [39]. Inspired by the hierarchical structure of fluorescent proteins in jellyfish and biomembranes in nature, they devised a universal strategy for constructing such hydrogels. Specifically, these hydrogels were created through the aqueous self-assembly and polymerization of UPy-containing polyelectrolyte-surfactant (i.e., SDS) micelles featuring a hydrophobic core with reversible physical crosslinks that could provide a durable molecular network structure. As a result, enhanced fluorescence emission was achieved through the formation of nanoclusters with electron-rich moieties that restrict intramolecular motion via hydrogen bonding networks. These hydrogels also exhibited bending behavior, which could be explained by the Shiga-type bending theory, where immobilized sulfonate ions in the hydrogel and freely moving Na^+^ ions migrated toward the cathode under the electric field. This migration could create an ionic strength difference, leading to an osmotic pressure difference that induces bending. Collectively, this type of micelle-containing hydrogel has potential applications in soft robotics, artificial muscles, and other bioelectronics.

## 4. Micelle-Containing Hydrogels as Delivery Systems

One of the most important directions of biomedical engineering might be controlled drug release for disease therapeutics [69]. As we described before, traditional hydrogels are ideal tools for the sustained delivery of hydrophilic drugs due to their unique molecular networks, highly hydrophilic matrices, and outstanding biocompatibility. There are two types of microenvironments within the micelle-containing hydrogels, i.e., the hydrophilic environment of the hydrogel matrices and the hydrophobic environment within the micelles, which make micelle hydrogels more flexible in storing and delivering diverse therapeutic agents (Figure 4) [70].

Post-operative recurrence of breast cancer poses a significant clinical challenge, particularly for patients who have undergone breast-conserving therapy [71]. In 2015, the Qian group developed a biodegradable and thermoresponsive hybrid hydrogel to prevent such recurrences [72]. Specifically, this hydrogel system incorporated gold nanorods (GNRs) into a thermoresponsive matrix and utilized a near-infrared (NIR) laser to trigger the release of loaded doxorubicin (DOX) through the photothermal effect of GNRs. In a mouse model of breast cancer recurrence, the DOX-PCNA-GNR hydrogel with NIR irradiation significantly reduced tumor recurrence to 16.7%, compared to the higher recurrence rates in the control groups. The researchers also observed that under 808 nm NIR laser irradiation, the hydrogel temperature rapidly increased to 50 °C, effectively killing cancer cells. This temperature change caused a significant contraction of the hydrogel and thus accelerated DOX release. Notably, this hydrogel demonstrated excellent thermal responsiveness and could be reused multiple times. Moreover, the sustained release of DOX could be significantly increased upon NIR laser irradiation or under acidic conditions (such as the tumor microenvironment), showcasing a remarkable characteristic for dual-stimuli-responsive drug release as a smart material.

In 2011, the Yang group reported a pH-triggered injectable amphiphilic hydrogel [40], which was synthesized through a Schiff’s base reaction between glycol chitosan (GC) and benzaldehyde-capped poly(ethylene glycol)-block-poly(propylene glycol)-block-poly(ethylene glycol) (OHC-PEO-PPO-PEO-CHO). The inclusion of DOX accelerated the gelation time and increased the gel strength, while paclitaxel (PTX) had the opposite effect. The release rates of DOX and PTX from this hydrogel system varied with the pH, where a lower pH accelerated the drug release. Future in vivo experiments showed that the hydrogel significantly reduced the DOX concentration in the bloodstream, thereby mitigating DOX-induced cardiotoxicity. Local injection of this hydrogel into subcutaneous tumors in mice showed an effective inhibition of tumor growth. Specifically, following the subcutaneous (s.c.) injection of free DOX, the peak blood concentration (C_max_) reached 2.2 µg/mL, which then rapidly declined within 80 h. In contrast, mice administered the DOX-loaded hydrogel exhibited a much lower C_max_ of approximately 0.05 µg/mL. Notably, the combination therapy with both drugs in the hydrogel significantly improved the survival time compared to single-drug therapies. Overall, these results indicated that this micelle-containing hydrogel could effectively control the drug release in a sustained manner and enhance the safety and efficacy of localized tumor therapies.

In 2009, the Muhammed group designed and developed a type of injectable superparamagnetic ferrogel (SPEL) for the controlled release of hydrophobic drugs (Figure 3D) [41]. These ferrogels were created using superparamagnetic iron oxide nanoparticles (SPIONs) embedded in Pluronic F127 (PF127) copolymer gels. The resulting hydrogels demonstrated temperature-dependent sol–gel transitions and magnetic field-responsive properties for drug release, which were ideal for targeted topical drug delivery. Specifically, SPIONs were mixed with a PF127 aqueous solution at low temperatures to form a stable suspension. Upon heating, PF127 micelles organized into an ordered structure, encapsulating the SPIONs. Thereafter, indomethacin (IMC), a model hydrophobic drug, was incorporated into the hydrophobic cores of PF127 micelles. Without a magnetic field, the drug release from this hydrogel system occurred primarily through diffusion. When an external magnetic field was applied, SPIONs aligned and moved closer, which could cause the micelles to “squeeze” and increase the local concentration gradient of the IMC, thus accelerating the drug release. Notably, the release half-time of the IMC decreased from 3195 min in the absence of a magnetic field to 1500 min in the presence of a 300 mT magnetic field.

Controlled release technologies are essential for tissue and immuno-engineering but often rely on passive diffusion, which largely limits the drug size and hampers the coordination of multiple drug releases [73]. In 2022, the Appel Lab introduced a novel injectable liposome-based supramolecular hydrogel (LNH) for the programmable release of multiple protein drugs [42]. Created by mixing liposomes with modified cellulose polymers, LNHs feature tunable mechanical properties and easy injectability through standard needles. By chemically engineering the liposome surfaces, these hydrogels enabled an affinity-based protein release, acting as depots for synchronized therapeutic delivery. In vivo studies in mouse models showed that LNHs could achieve a sustained and localized release of IgG antibodies as well as IL-12 cytokines despite their size differences. The in vivo imaging system (IVIS) indicated that while bolus administration caused a swift reduction in both IgG and IL-12 within a day, these liposomal micelle hydrogels synchronized the release patterns of these proteins, retaining over 50% of the fluorescent signal even after 14 days. The follow-up toxicology studies showed that the gradual degradation of LNHs in vivo would not cause toxicity or inflammation. The two types of supramolecular forces within this hydrogel network, i.e., electrostatic and hydrophobic interactions, effectively reduced the release rates of protein cargos, which made the liposomal micelle hydrogel an ideal smart material for sustained drug delivery.

## 5. Summary and Outlooks

As one of the most commonly used materials in our daily lives, hydrogels are now becoming more and more important in biomedical research and clinics (Figure 1) [74]. Among the various types of hydrogels, micelle-containing hydrogels possess significant advantages (Table 1), which can effectively address the challenge of delivering poorly water-soluble drugs. Their unique amphiphilic micelle structures allow for the efficient encapsulation of water-insoluble drugs, thereby facilitating effective drug delivery in a sustained manner. As smart materials, micelle-containing hydrogels offer unparalleled advantages in drug delivery systems. Specifically, they are able to respond to various environmental stimuli (Figure 3), such as changes in pH, temperature, magnetic fields, electric fields, enzymes, metal ions, irradiation, etc., enabling a precise drug release in different physiological environments and the regulation of release rates (Figure 4). Additionally, in comparison with other materials, hydrogels possess excellent biocompatibility and degradability, making them a safe and effective carrier for biomedical applications. In summary, well-designed micelle-containing hydrogels possess unique internal matrix structures with dual microenvironments, which enable them to have novel material properties and great application prospects in the biomedical field.

## Figures and Tables

**Figure 1 gels-10-00471-f001:**
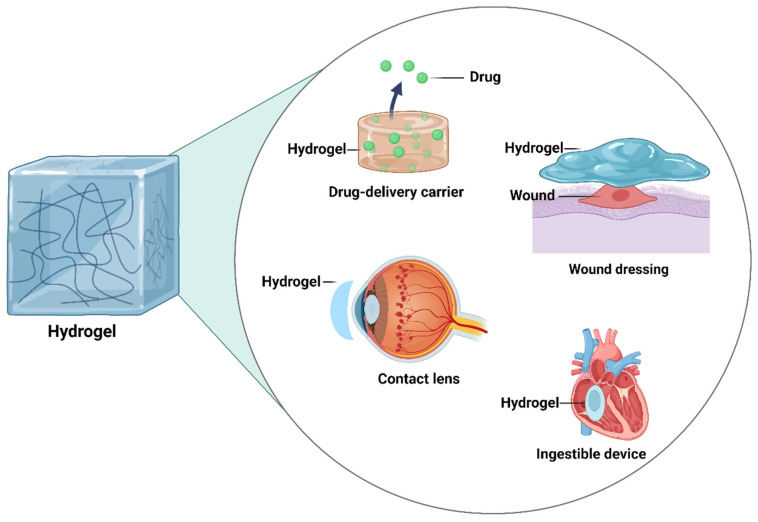
Representative biomedical applications of hydrogels, which are one of the most commonly used materials in our daily lives.

**Figure 2 gels-10-00471-f002:**
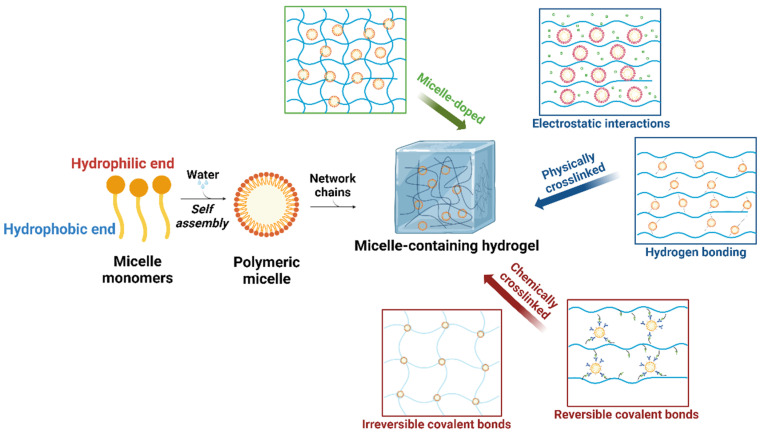
The formation process of micelle-containing hydrogels and their two main types, i.e., micelle-doped and micelle-crosslinked hydrogels.

**Figure 3 gels-10-00471-f003:**
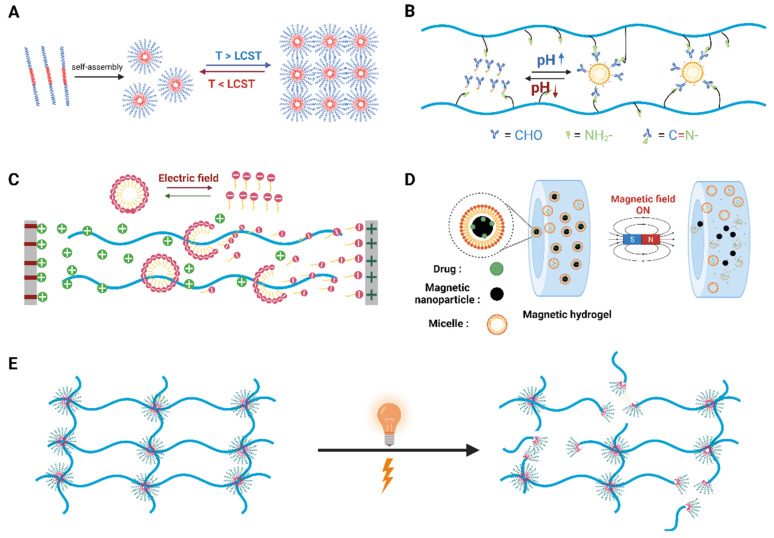
Schematic diagram showing the responses of micelle-containing hydrogels to representative environmental stimuli, including changes in temperature (**A**), pH (**B**), electric fields (**C**), magnetic fields (**D**), and irradiation (**E**).

**Figure 4 gels-10-00471-f004:**
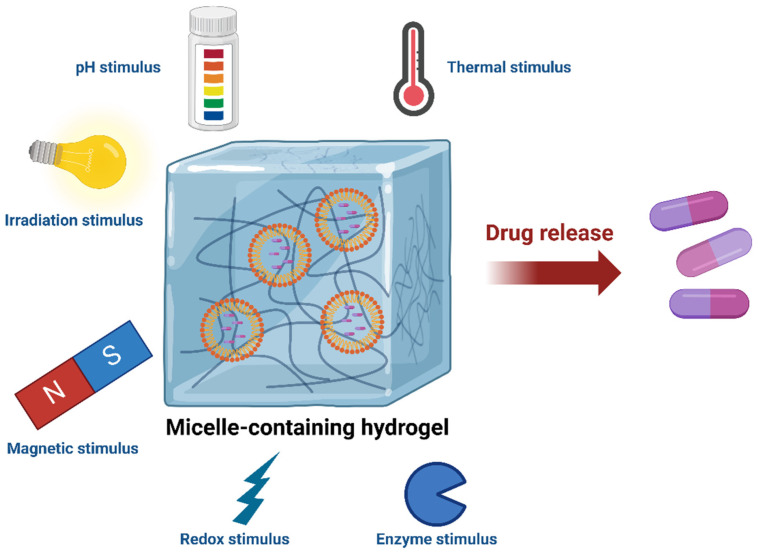
Summary and application perspective of micelle-containing hydrogels in biomedical research, especially for controlled drug release.

**Table 1 gels-10-00471-t001:** Summary of the micelle-containing hydrogel examples described in this paper.

Micelle Formulation	Category	Application	Reference
Resveratrol	Doped	Drug release	Joseph, A. et al., 2022 [29]
PIPAAm	Doped	Thermo sensor	Yoshida, R. et al., 1995 [30]
Cholesterol-hyaluronate	Doped	Contact lens	Mun, J. et al., 2019 [6]
Modified PEG	Chemically crosslinked	Drug delivery	Sawhney, A. S.; et al., 1993 [17]
PU	Chemically crosslinked	Memory material	Tan, M. et al., [31,32]
UPy	Chemically crosslinked	Cell encapsulation	Qin, Z. et al., 2019 [33]
Pluronic F127	Physically crosslinked	Memory material	Yang, H. et al., 2022 [34]
CDAB	Physically crosslinked	Strain sensor	Khan, M. et al., 2023 [35]
PEG-PLA	Physically crosslinked	Lacrimal implant	Zhao, J. et al., 2023 [36]
SDS	Physically crosslinked	Gas therapy	Zhou, X. et al., 2023 [37]
SDS	Physically crosslinked	Drug release	Zhou, X. et al., 2023 [38]
SDS	Physically crosslinked	Soft robotics	Tao, L. et al., 2021 [39]
PEO-PPO-PEO	Chemically crosslinked	Drug delivery	Zhao, L. et al., 2011 [40]
F127 and SPION	Physically crosslinked	Drug release	Qin, J. et al., 2009 [41]
Liposome	Physically crosslinked	Drug release	Correa, S. et al., 2022 [42]

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
