# Peer review of "Micelle-Containing Hydrogels and Their Applications in Biomedical Research"

_gels, 2024, doi:10.3390/gels10070471_

Round 1
Reviewer 1 Report
Comments and Suggestions for Authors
Current review article focus on the application of micelle loaded hydrogel for the biomedical research. Hydrogel is the one of the hot topic recently due to their unique properties of controlling drug release and biodegradability. Author try to cover many application relevant to micelle but still is wide topic.
Overall, article is written well and justified as much possible but still need improvement.
1) Is it possible then cover the specific area such as cancer, wound and other disease and in that described available micelle loaded hydrogel formulation.
2) In introduction section, formulation part is written down but not found in the article
3) Kindly draw diagram of formulation of micelle and also hydrogel
4) Highlight the what is drawback of micelle alone and which overcome using hydrogel
5) Make a table of list of micelle loaded hydrogel and their application
6) If any marketed formulation available then add in manuscript.
Author Response
Reviewer 1:
Current review article focus on the application of micelle loaded hydrogel for the biomedical research. Hydrogel is the one of the hot topic recently due to their unique properties of controlling drug release and biodegradability. Author try to cover many application relevant to micelle but still is wide topic.
Overall, article is written well and justified as much possible but still need improvement.
1) Is it possible then cover the specific area such as cancer, wound and other disease and in that described available micelle loaded hydrogel formulation.
Thank you for the advice. In this short review paper, we focused on introducing micelle-containing hydrogel types and their representative applications in biomedical research. Please note that this is not a comprehensive review collecting the details. More importantly, as an emerging hydrogel material, up to now micelle-containing hydrogels have not been employed in many kinds of diseases. For more details of the corresponding applications, please see the newly added Table 1.
2) In introduction section, formulation part is written down but not found in the article
The formulation of micelles has been described in the main text and now also been added to Table 1.
3) Kindly draw diagram of formulation of micelle and also hydrogel
To make this paper more readable and have a broad readership, we chose to use cartoons to stand for the micelles and polymer chains within the hydrogels. The formulation of micelles has been described in the main text and now also been added to Table 1.
4) Highlight the what is drawback of micelle alone and which overcome using hydrogel
Thank you for your suggestion. We have added the descriptions accordingly in Section 2, which now reads:
“Due to the instability, low local concentrations, and other issues of micelles in complex biological environments, directly using micelles as carriers for the sustained delivery of therapeutic agents still stays challenging [29], while loading micelles into the hydrogels can effectively address these issues for a broader application range in biomedical research and clinic.”
5) Make a table of list of micelle loaded hydrogel and their application
Please see the newly added Table 1.
|
Micelle formulation |
Category |
Application |
Reference |
|
Resveratrol |
Doped |
Drug release |
Joseph, A.; et al., 2022 [32] |
|
PIPAAm |
Doped |
Thermo sensor |
Yoshida, R.; et al., 1995 [34] |
|
Cholesterol-hyaluronate |
Doped |
Contact lens |
Mun, J.; et al., 2019 [6] |
|
Modified PEG |
Chemically crosslinked |
Drug delivery |
Sawhney, A. S.; et al., 1993 [17] |
|
PU |
Chemically crosslinked |
Memory material |
Tan, M.; et al., [43,44]. |
|
UPy |
Chemically crosslinked |
Cell encapsulation |
Qin, Z. et al., 2019 [49] |
|
Pluronic F127 |
Physically crosslinked |
Memory material |
Yang, H.; et al., 2022[54] |
|
CDAB |
Physically crosslinked |
Strain sensor |
Khan, M.; et al., 2023 [55] |
|
PEG-PLA |
Physically crosslinked |
Lacrimal implant |
Zhao, J.; et al., 2023 [56] |
|
SDS |
Physically crosslinked |
Gas therapy |
Zhou, X.; et al., 2023 [63] |
|
SDS |
Physically crosslinked |
Drug release |
Zhou, X.; et al., 2023 [64] |
|
SDS |
Physically crosslinked |
Soft robotics |
Tao, L.; et al., 2021 [65] |
|
PEO-PPO-PEO |
Chemically crosslinked |
Drug delivery |
Zhao, L.; et al., 2011 [70] |
|
F127 and SPION |
Physically crosslinked |
Drug release |
Qin, J.; et al., 2009 [71] |
|
Liposome |
Physically crosslinked |
Drug release |
Correa, S.; et al., 2022 [73] |
6) If any marketed formulation available then add in manuscript.
The information has been added accordingly to the main text as well as Table 1.
Reviewer 2 Report
Comments and Suggestions for Authors
Dear Author,
Your manuscript entitled “Micelle-containing Hydrogels and Their Applications in Biomedical Research” has been comprehensively evaluated. Although the manuscript was well designed and the topic was wisely selected, there was a need for a generation of better manuscript. Thus, minor revisions were requested to address some of the conflicts. Hereby, I would like to present my comments.
1. Please give information about the selection of the studies in your manuscript. Which databases were scanned? Was there a specific period scanned (e.g. last 20 years?) What was inclusion or exclusion criteria when selecting the literature data? Please clarify this issue.
2. 2.2.1. Chemically crosslinked micelle hydrogels section, line 157: …hydrogels with unique properties [40].” Please give examples about “the unique properties” (such as controlled release, pH stimulated release etc.).
3. Grammar should be checked meticulously. (e.g. line 208: …was report recently.)
4. 2.2.2. Physically crosslinked micelle hydrogels section, line 227: F127? Is it Pluronic 127? please give at least one explanation in the first place of usage.
5. (As a suggestion) 4. Micelle-containing hydrogels as delivery systems section: Please make a comprehensive literature search based on the pharmacological action or route of application (anticancer activity or ophthalmic route). Then, please share your findings in a summarized table.
6. The references section should be carefully monitored. The journal names should be abbreviated as indicated in the instructions for authors. (e.g. REF11,12,19,29,52,66,70)
Best regards.

Author Response
Reviewer 2:
Your manuscript entitled “Micelle-containing Hydrogels and Their Applications in Biomedical Research” has been comprehensively evaluated. Although the manuscript was well designed and the topic was wisely selected, there was a need for a generation of better manuscript. Thus, minor revisions were requested to address some of the conflicts. Hereby, I would like to present my comments.
- Please give information about the selection of the studies in your manuscript. Which databases were scanned? Was there a specific period scanned (e.g. last 20 years?) What was inclusion or exclusion criteria when selecting the literature data? Please clarify this issue.
Thank you for the comments. The literature we cited is selected based on the content, and thus there was no specific timeline limit or database restriction. Briefly, we chose representative and topic-related studies from mainstream journals. In this short review, we focused on the introduction of micelle-containing hydrogel types and their representative applications in biomedical research. Please note that this is not a comprehensive review collecting the details or meta-analysis.
- 2.2.1. Chemically crosslinked micelle hydrogels section, line 157: …hydrogels with unique properties [40].” Please give examples about “the unique properties” (such as controlled release, pH stimulated release etc.).
The statement has been revised accordingly, which now reads:
“This technique has been frequently used to prepare tough hydrogels with unique mechanical properties [42].”
- Grammar should be checked meticulously. (e.g. line 208: …was report recently.)
Thank you for your careful reading. We have made the revisions accordingly.
- 2.2.2. Physically crosslinked micelle hydrogels section, line 227: F127? Is it Pluronic 127? please give at least one explanation in the first place of usage.
Thank you for the comments. We have made the revisions accordingly, which now reads:
“Another example is Pluronic F127 diacrylate (F127DA)-based micelle hydrogels, which was synthesized by acrylating the end hydroxyl groups of Pluronic F127 using acryloyl chloride and triethylamine.”
- (As a suggestion) 4. Micelle-containing hydrogels as delivery systems section: Please make a comprehensive literature search based on the pharmacological action or route of application (anticancer activity or ophthalmic route). Then, please share your findings in a summarized table.
Thank you for the suggestion. Please see the newly added Table 1.
- The references section should be carefully monitored. The journal names should be abbreviated as indicated in the instructions for authors. (e.g. REF11,12,19,29,52,66,70)
Thank you for the careful reading. The citations have been revised accordingly.
Round 2
Reviewer 1 Report
Comments and Suggestions for Authors
The revision is satisfactory.